# Aligning individual brains
# with Fused Unbalanced Gromov-Wasserstein

**Alexis Thual**
Cognitive Neuroimaging Unit, INSERM, CEA, CNRS, NeuroSpin center, Gif sur Yvette, France
Mind, Inria Paris-Saclay, Palaiseau, France
Inserm, Collège de France, Paris, France
alexis.thual@cea.fr

**Huy Tran**
CMAP, Ecole Polytechnique, Palaiseau, France
Université Bretagne-Sud, CNRS, IRISA, Vannes, France
quang-huy.tran@univ-ubs.fr

**Tatiana Zemskova**
Mind, Inria Paris-Saclay, Palaiseau, France
tatiana.zemskova@polytechnique.edu

**Nicolas Courty**
Université Bretagne-Sud, CNRS, IRISA, Vannes, France
ncourty@irisa.fr

**Rémi Flamary**
CMAP, Ecole Polytechnique, Palaiseau, France
remi.flamary@polytechnique.edu

**Stanislas Dehaene**
Cognitive Neuroimaging Unit, INSERM, CEA, CNRS, NeuroSpin center, Gif sur Yvette, France
Inserm, Collège de France, Paris, France
stanislas.dehaene@cea.fr

**Bertrand Thirion**
Mind, Inria Paris-Saclay, Palaiseau, France
bertrand.thirion@inria.fr

36th Conference on Neural Information Processing Systems (NeurIPS 2022).

## Abstract

Individual brains vary in both anatomy and functional organization, even within a given species. Inter-individual variability is a major impediment when trying to draw generalizable conclusions from neuroimaging data collected on groups of subjects. Current co-registration procedures rely on limited data, and thus lead to very coarse inter-subject alignments. In this work, we present a novel method for inter-subject alignment based on Optimal Transport, denoted as Fused Unbalanced Gromov Wasserstein (FUGW). The method aligns cortical surfaces based on the similarity of their functional signatures in response to a variety of stimulation settings, while penalizing large deformations of individual topographic organization. We demonstrate that FUGW is well-suited for whole-brain landmark-free alignment. The unbalanced feature allows to deal with the fact that functional areas vary in size across subjects. Our results show that FUGW alignment significantly increases between-subject correlation of activity for independent functional data, and leads to more precise mapping at the group level.

## 1 Introduction

The availability of millimeter or sub-millimeter anatomical or functional brain images has opened new horizons to neuroscience, namely that of mapping cognition in the human brain and detecting markers of diseases. Yet this endeavour has stumbled on the roadblock of inter-individual variability: while the overall organization of the human brain is largely invariant, two different brains (even from monozygotic twins [33]) may differ at the scale of centimeters in shape, folding pattern, and functional responses. The problem is further complicated by the fact that functional images are noisy, due to imaging limitations and behavioral differences across individuals that cannot be easily overcome. The status quo of the field is thus to rely on anatomy-based inter-individual alignment that approximately matches the outline of the brain [4] as well as its large-scale cortical folding patterns [12, 15]. Existing algorithms thus coarsely match anatomical features with diffeomorphic transformations, by warping individual data to a simplified template brain. Such methods lose much of the original individual detail and blur the functional information that can be measured in brain regions (see Figure 1).

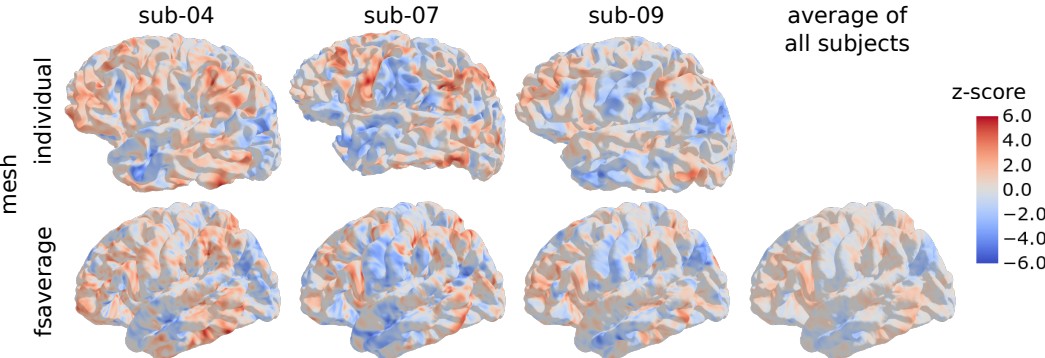

**Figure 1: High variability in human anatomies and functional MRI responses across subjects** In this experiment contrasting areas of the brain which respond to mathematical tasks against other that don't, we observe great variability in locations and strength of brain activations across subjects (row 1). The classical approach consists in wrapping this data to a common surface template (row 2), where they can be averaged, often resulting in loss of individual details and detection power. These images were generated using Nilearn software [1].

In order to improve upon the current situation, a number of challenges have to be addressed: *(i)* There exists no template brain with functional information, which by construction renders any cortical matching method blind to function. This is unfortunate, since functional information is arguably the most accessible marker to identify cortical regions and their boundaries [18]. *(ii)* When comparing two brains – coming from individuals or from a template – it is unclear what regularity should be

imposed on the matching [42]. While it is traditional in medical imaging to impose diffeomorphicity [4], such a constrain does not match the frequent observation that brain regions vary across individuals in their fine-grained functional organization [18, 38]. *(iii)* Beyond the problem of aligning human brains, it is an even greater challenge to systematically compare functional brain organization in two different species, such as humans and macaques [26, 23, 46, 14]. Such inter-species comparisons introduce a more extreme form of variability in the correspondence model.

**Related work** Several attempts have been made to constrain the brain alignment process by using functional information. The first one consists in introducing functional maps into the diffeomorphic framework and search for a smooth transformation that matches functional information [37, 47, 35], the most popular framework being arguably Multimodal Surface Matching (MSM) [35, 18].

A second family of less constrained functional alignment approaches have been proposed, based on heuristics, by matching information in small, possibly overlapping, cortical patches [21, 40, 6]. This popular framework has been called *hyperalignment* [21, 20], or *shared response models* [10]. Yet these approaches lack a principled framework and cannot be considered to solve the matching problem at scale. Neither do they allow to estimate a group-level template properly [45].

An alternative functional alignment framework has followed another path [19], considering functional signal as a three-dimensional distribution, and minimizing the transport cost. However, this framework imposes unnatural constraints of non-negativity of the signal and only works for one-dimensional contrasts, so that it cannot be used to learn multi-dimensional anatomo-functional structures. An important limitation of the latter two families of methods is that they operate on a fixed spatial context (mesh or voxel grid), and thus cannot be used on heterogeneous meshes such as between two individual human anatomies or, worse, between a monkey brain and a human brain.

**Contributions** Following [5], we use the Wasserstein distance between source and target functional signals – consisting of contrast maps acquired with fMRI – to compute brain alignments. We contribute two notable extensions of this framework: *(i)* a Gromov-Wasserstein (GW) term to preserve global anatomical structure – this term introduces an anatomical penalization against improbably distant anatomical matches, yet without imposing diffeomorphic regularity – as well as *(ii)* an unbalanced correspondence that allows mappings from one brain to another to be incomplete, for instance because some functional areas are larger in some individuals than in others, or may simply be absent. We show that this approach successfully addresses the challenging case of different cortical meshes, and that derived brain activity templates are sharper than those obtained with standard anatomical alignment approaches.

## 2    Methods

Optimal Transport yields a natural framework to address the alignment problem, as it seeks to derive a plan – a *coupling* – that can be seen as a soft assignment matrix between cortical areas of a source and target individual. As discussed previously, there is a need for a functional alignment method that respects the rich geometric structure of the anatomical features, hence the Wasserstein distance alone is not sufficient. By construction, the GW distance [24, 25] can help preserve the global geometry underlying the signal. The more recent fused GW distance [44] goes one step further by making it possible to integrate functional data simultaneously with anatomical information.

### 2.1    Fused Unbalanced Gromov-Wasserstein

We leverage [44, 39] to present a new objective function which interpolates between a loss preserving the global geometry of the underlying mesh structure and a loss aligning source and target features, while simultaneously allowing not to transport some parts of the source and target distributions. We provide an open-source solver that minimizes this loss[1].

**Formulation** We denote $\boldsymbol{F^s} \in \mathbb{R}^{n,c}$ the matrix of features per vertex for the source subject. In the proposed application, they correspond to $c$ functional activation maps, sampled on a mesh with $n$ vertices representing the source subject's cortical surface. Let $\boldsymbol{D^s} \in \mathbb{R}^{n,n}_+$ be the matrix of pairwise

---

[1]https://github.com/alexisthual/fugw provides a PyTorch [28] solver with a scikit-learn [29] compatible API

geodesic distances[2] between vertices of the source mesh. Moreover, we assign the distribution $\boldsymbol{w^s} \in \mathbb{R}^n_+$ on the source vertices. Comparably, we define $\boldsymbol{F^t} \in \mathbb{R}^{p,c}$, $\boldsymbol{D^t} \in \mathbb{R}^{p,p}_+$ and $\boldsymbol{w^t} \in \mathbb{R}^p_+$ for the target subject, whose individual anatomy is represented by a mesh comprising $p$ vertices. Eventually, $\boldsymbol{w^s}$ and $\boldsymbol{w^t}$ set the transportable mass per vertex, which, without prior knowledge, we choose to be uniform for the source and target vertices respectively: $\boldsymbol{w^s} \triangleq (\frac{1}{n}, ..., \frac{1}{n})$, $\boldsymbol{w^t} \triangleq (\frac{1}{p}, ..., \frac{1}{p})$.

Given a tuple of hyper-parameters $\theta \triangleq (\rho, \alpha, \varepsilon)$, where $\rho, \varepsilon \in \mathbb{R}_+$ and $\alpha \in [0, 1]$, for any coupling $\boldsymbol{P} \in \mathbb{R}^{n,p}$, we define the fused unbalanced Gromov-Wasserstein loss as

$$
\begin{aligned}
\mathrm{L}_\theta(\boldsymbol{P}) \triangleq (1-\alpha) \underbrace{\sum_{\substack{0 \le i < n \\ 0 \le j < p}} ||\boldsymbol{F^s_i} - \boldsymbol{F^t_j}||^2_2 \boldsymbol{P}_{i,j}}_{\text{Wasserstein loss } \mathrm{L_W}(\boldsymbol{P})} + \alpha \underbrace{\sum_{\substack{0 \le i,k < n \\ 0 \le j,l < p}} |\boldsymbol{D^s_{i,k}} - \boldsymbol{D^t_{j,l}}|^2 \boldsymbol{P}_{i,j} \boldsymbol{P}_{k,l}}_{\text{Gromov-Wasserstein loss } \mathrm{L_{GW}}(\boldsymbol{P})} \\
+ \rho \big( \underbrace{\mathrm{KL}(\boldsymbol{P}_{\#1} \otimes \boldsymbol{P}_{\#1} | \boldsymbol{w^s} \otimes \boldsymbol{w^s}) + \mathrm{KL}(\boldsymbol{P}_{\#2} \otimes \boldsymbol{P}_{\#2} | \boldsymbol{w^t} \otimes \boldsymbol{w^t})}_{\text{Marginal constraints } \mathrm{L_U}(\boldsymbol{P})} \big) + \varepsilon \underbrace{E(\boldsymbol{P})}_{\text{Entropy}}
\end{aligned}
\tag{1}
$$

where $\mathrm{L_W}(\boldsymbol{P})$ matches vertices with similar features, $\mathrm{L_{GW}}(\boldsymbol{P})$ penalizes changes in geometry and $\mathrm{L_U}(\boldsymbol{P})$ fosters matching all parts of the source and target distributions. Throughout this paper, we refer to relaxing the hard marginal constraints of the underlying OT problem into soft ones as *unbalancing*. Here, $\boldsymbol{P}_{\#1} \triangleq (\sum_j \boldsymbol{P}_{i,j})_{0 \le i < n}$ denotes the first marginal distribution of $\boldsymbol{P}$, and $\boldsymbol{P}_{\#2} \triangleq (\sum_i \boldsymbol{P}_{i,j})_{0 \le j < p}$ the second marginal distribution of $\boldsymbol{P}$. The notation $\otimes$ represents the Kronecker product between two vectors or two matrices. $\mathrm{KL}(\cdot|\cdot)$ denotes the Kullback Leibler divergence, which is a typical choice to measure the discrepancy between two measures in the context of unbalanced optimal transport [22]. The last term $E(\boldsymbol{P}) \triangleq \mathrm{KL}\big(\boldsymbol{P} \otimes \boldsymbol{P} | (\boldsymbol{w^s} \otimes \boldsymbol{w^t}) \otimes (\boldsymbol{w^s} \otimes \boldsymbol{w^t})\big)$ is mainly introduced for computational purposes, as it helps accelerate the approximation scheme of the optimisation problem. Typically, it is used in combination with a small value of $\varepsilon$, so that the impact of other terms is not diluted. On the other hand, the parameters $\alpha$ and $\rho$ offer control over two other aspects of the problem: while $\alpha$ realizes a trade-off between the impact of different features and different geometries in the resulting alignment, $\rho$ controls the amount of mass transported by penalizing configurations such that the marginal distributions of the transportation plan $\boldsymbol{P}$ are far from the prior weights $\boldsymbol{w^s}$ and $\boldsymbol{w^t}$. This potentially helps adapting the size of areas where either the signal or the geometry differs too much between source and target.

Eventually, we define $\boldsymbol{\mathcal{X}^s} \triangleq (\boldsymbol{F^s}, \boldsymbol{D^s}, \boldsymbol{w^s})$ and $\boldsymbol{\mathcal{X}^t} \triangleq (\boldsymbol{F^t}, \boldsymbol{D^t}, \boldsymbol{w^t})$, and seek to derive an optimal coupling $\boldsymbol{P} \in \mathbb{R}^{n,p}$ minimizing

$$
\mathrm{FUGW}(\boldsymbol{\mathcal{X}^s}, \boldsymbol{\mathcal{X}^t}) \triangleq \inf_{\boldsymbol{P} \ge 0} L_\theta(\boldsymbol{P})
\tag{2}
$$

This can be seen as a natural combination of the fused GW [44] and the unbalanced GW [39] distances. To the best of our knowledge, it has never been considered in the literature. Following [39], we approximate FUGW via a lower bound. It involves solving a minimization problem with respect to two independent couplings: Using a Block-Coordinate Descent (BCD) scheme, we fix a coupling and minimize with respect to the other. This allows us to always be dealing with linear problems instead of a quadratic one. Eventually, each BCD iteration consists in alternatively solving two entropic unbalanced OT problems, whose solutions can be approximated using the scaling algorithm [11]. Details concerning the lower bound as well as the corresponding BCD iteration can be found in the Appendix (see Alg. S1).

**Toy example illustrating the unbalancing property** As exemplified in Figure 1, brain responses elicited by the same stimulus vary greatly between individuals. Figure 2 illustrates a similar yet simplified version of this problem, where the goal is to align two different signals supported on the same spherical meshes. In this example, for each of the $n = p = 3200$ vertices, the feature is simply a scalar. On the source mesh, the signal is constituted of two von Mises density functions that differ by their concentration (large and small), while on the target mesh, only the large one is present, but at a different location. We use the optimal coupling matrix $\boldsymbol{P}$ obtained from Eq. 2 to transport the source signal on the target mesh. As shown in Figure 2.B, the parameter $\rho$ allows to control the mass

---

[2]We compute geodesic distances using https://github.com/the-virtual-brain/tvb-gdist

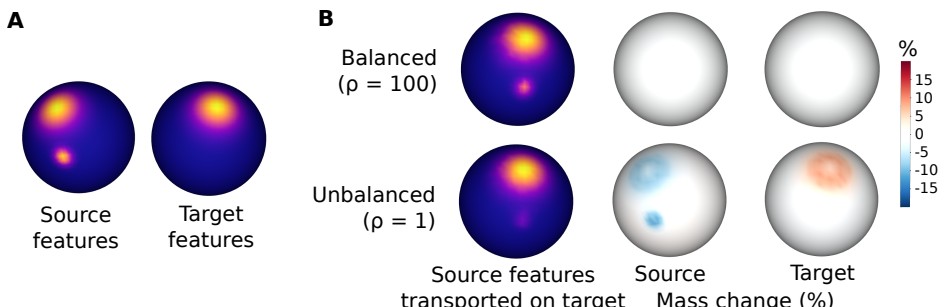

**Figure 2: Unbalancing helps accounting for idiosyncrasies of the source and target signals**
When trying to align the source and target signals (Panel A), the classical balanced setup (Panel B, top row) transports all parts of the source signal even if they have no counterpart in the target signal. In the unbalanced setup (Panel B, bottom row), less source-only signal is transported: in particular, less mass is transported from the source's small blob onto the target (Panel B, middle column).

transferred from source to target. When $\rho = 100$, we approach the solution of the fused GW problem. Consequently, we observe the second mode on the target when transporting the source signal. When the mass control is weaker ($\rho = 1$), the smaller blob is partly removed because it has no counterpart in the target configuration, making the transport ill-posed.

**Barycenters** Barycenters represent common patterns across samples. Their role is instrumental in identifying a unique target for aligning a given group of individuals. As seen in Fig. 1, the vertex-wise group average does not usually provide well-contrasted maps. Inspired by the success of the GW distance when estimating the barycenter of structured objects [30, 44], we use FUGW to find the barycenter $(\boldsymbol{F}^B, \boldsymbol{D}^B) \in \mathbb{R}^{k,c} \times \mathbb{R}^{k,k}$ of all subjects $s \in \mathcal{S}$, as well as the corresponding couplings $\boldsymbol{P}^{s,B}$ from each subject to the barycenter. More precisely, we solve

$$\boldsymbol{\mathcal{X}}^B = (\boldsymbol{F}^B, \boldsymbol{D}^B, \boldsymbol{w}^B) \in \arg\min_{\boldsymbol{\mathcal{X}}} \sum_{s \in \mathcal{S}} \text{FUGW}(\boldsymbol{\mathcal{X}}^s, \boldsymbol{\mathcal{X}}), \tag{3}$$

where we set the weights $\boldsymbol{w}_B$ to be the uniform distribution. By construction, the resulting barycenter benefits from the advantages of FUGW, i.e. equilibrium between geometry-preserving and feature-matching properties, while not forcing hard marginal constraints. The FUGW barycenter is estimated using a Block-Coordinate Descent (BCD) algorithm that consists in alternatively *(i)* minimizing the OT plans $\boldsymbol{P}^{s,B}$ for each FUGW computation in (3) with fixed $\boldsymbol{\mathcal{X}}^B$ and *(ii)* updating the barycenter $\boldsymbol{\mathcal{X}}^B$ through a closed form with fixed $\boldsymbol{P}^{s,B}$. See Alg. S4 for more details.

## 3 Numerical experiments

We design three experiments to assess the performance of FUGW. In Experiments 1 and 2, we are interested in assessing if aligning pairs of individuals with FUGW increases correlation between subjects compared to a baseline correlation. We also compare the ensuing gains with those obtained when using the competing method MSM [35, 36] to align subjects. In Experiment 3, we derive a barycenter of individuals and assess its ability to capture fine-grained details compared to classical methods.

**Dataset** In all three experiments, we leverage data from the Individual Brain Charting dataset [31]. It is a longitudinal study on 12 human subjects, comprising 400 fMRI maps per subject collected on a wide variety of stimuli (motor, visual, auditory, theory of mind, language, mathematics, emotions, and more), movie-watching data, T1-weighted maps, as well as other features such as retinotopy which we don't use in this work. We leverage these 400 fMRI maps. The training, validation and test sets respectively comprise 326, 43 and 30 contrast maps acquired for each individual of the dataset. Tasks and MRI sessions differ between each of the sets. More details, including preprocessing, are provided in Supplementary Materials.

**Baseline alignment correlation**   For each pair of individuals $(s, t)$ under study, and for each fMRI contrast $c$ in the test set, we compute the Pearson correlation $\text{corr}(\boldsymbol{F}^s_{\cdot,c}, \boldsymbol{F}^t_{\cdot,c})$ after these maps have been projected onto a common surface anatomy (in this case, *fsaverage5* mesh). Throughout this work, such computations are made for each hemisphere separately.

**Experiment 1 - Aligning pairs of humans with the same anatomy**   For each pair $(s, t)$ under study, we derive an alignment $\boldsymbol{P}^{s,t} \in \mathbb{R}^{n \times p}$ using FUGW on a set of training features. In this experiment, source and target data lie on the same anatomical mesh (*fsaverage5*), and $n = p = 10\,240$ for each hemisphere. Since each hemisphere's mesh is connected, we align one hemisphere at a time.

Computed couplings are used to align contrast maps of a the validation set from the source subject onto the target subject. Indeed, one can define $\phi_{s \to t} \colon \boldsymbol{X} \in \mathbb{R}^{n \times q} \mapsto \left((\boldsymbol{P}^{s,t})^T \boldsymbol{X}\right) \oslash \boldsymbol{P}^{s,t}_{\#2} \in \mathbb{R}^{p \times q}$ where $\oslash$ represents the element-wise division. $\phi_{s \to t}$ transports any matrix of features from the source mesh to the target mesh. We measure the Pearson correlation $\text{corr}\left(\phi_{s \to t}(\boldsymbol{F}^s), \boldsymbol{F}^t\right)$ between each aligned source and target maps.

We run a similar experiment for MSM and compute the correlation gain induced on a test set by FUGW and MSM respectively. For both models, we selected the hyper-parameters maximizing correlation gain on a validation set. In the case of FUGW, in addition to gains in correlation, hyper-parameter selection was influenced by three other metrics that help us assess the relevance of computed couplings:

**Transported mass**   For each vertex $i$ of the source subject, we compute $\sum\limits_{0 \le j < p} \boldsymbol{P}^{s,t}_{i,j}$

**Vertex displacement**   Taking advantage of the fact that the source and target anatomies are the same, we define $\boldsymbol{D} = \boldsymbol{D}^s = \boldsymbol{D}^t$ and compute for each vertex $i$ of the source subject the quantity $\sum_j \boldsymbol{P}^{s,t}_{i,j} \cdot \boldsymbol{D}_{i,j} / \sum_j \boldsymbol{P}^{s,t}_{i,j}$, which measures the average geodesic distance on the cortical sheet between vertex $i$ and the vertices of the target it has been matched with

**Vertex spread**   Large values of $\varepsilon$ increase the entropy of derived couplings. To quantify this effect, and because we don't want the matching to be too blurry, we assess how much a vertex was *spread*. Considering $\tilde{P}_i = \boldsymbol{P}^{s,t}_i / \sum_j \boldsymbol{P}^{s,t}_{i,j} \in \mathbb{R}^p$ as a probability measure on target vertices, we estimate the anatomical variance of this measure by sampling $q$ pairs $(j_q, k_q)$ of $\tilde{P}_i$ and computing their average geodesic distance $\frac{1}{q} \sum\limits_{j_q, k_q} \boldsymbol{D}_{j_q, k_q}$

**Experiment 2 - Aligning pairs of humans with individual anatomies**   We perform a second alignment experiment, this time using individual meshes instead of an anatomical template. Importantly, in this case, there is no possibility to compare FUGW with baseline methods, since those cannot handle this case.

However, individual meshes are significantly larger than the common anatomical template used in Experiment 1 ($n \approx m \approx$ 160k vs. 10k previously), resulting in couplings too large to fit on GPUs – for reference, a coupling of size 10k $\times$ 10k already weights  400Mo on disk. We thus reduce the size of the source and target data by clustering them into 10k small connected clusters using Ward's algorithm [41]. More details are given in supplementary section A.4.

**Experiment 3 - Comparing FUGW barycenters with usual group analysis**   Since it is very difficult to estimate the barycentric mesh, we force it to be equal to the *fsaverage5* template. Empirically, this we force the distance matrix $\boldsymbol{D}^B$ to be equal to that of *fsaverage5*, and only estimate the functional barycenter $\boldsymbol{F}^B$. We initialize it with the mean of $(\boldsymbol{F}^s)_{s \in S}$ and derive $\boldsymbol{F}^B$ and $(\boldsymbol{P}^{s,B})_{s \in S}$ from problem 3.

Then, for a given stimulus $c$, we compute its projection onto the barycenter for each subject. We use these projections to compute two maps of interest: *(i)* $\boldsymbol{M}_{B,c}$ the mean of projected contrast maps across subjects and *(ii)* $\boldsymbol{T}_{B,c}$ the t-statistic (for each vertex) of projected maps. We compare these two maps with their unaligned counterparts $\boldsymbol{M}_{0,c}$ and $\boldsymbol{T}_{0,c}$ respectively.

The first map helps us to qualitatively evaluate the precision of FUGW alignments and barycenter. The second one is classically used to infer the existence of areas of the brain that respond to specific

$$M_{B,c} \triangleq \frac{1}{|S|} \sum_{s \in S} \phi_{s \to t}(\boldsymbol{F}^{\boldsymbol{s}}_{\cdot,c}) \qquad \boldsymbol{T_{B,c}} \triangleq \text{t-statistic}\left(\left(\phi_{s \to t}(\boldsymbol{F}^{\boldsymbol{s}}_{\cdot,c})\right)_{s \in S}\right)$$

$$\boldsymbol{M_{0,c}} \triangleq \frac{1}{|S|} \sum_{s \in S} \boldsymbol{F}^{\boldsymbol{s}}_{\cdot,c} \qquad \boldsymbol{T_{0,c}} \triangleq \text{t-statistic}\left(\left(\boldsymbol{F}^{\boldsymbol{s}}_{\cdot,c}\right)_{s \in S}\right)$$

stimuli. We assess whether FUGW helps find the same clusters of vertices. Eventually, we quantify the number of vertices significantly activated or deactivated with and without alignment respectively.

## 4 Results

### 4.1 Experiment 1 - Template anatomy

**Aligning subjects on a fixed mesh** We set $\alpha = 0.5$, $\rho = 1$ and $\varepsilon = 10^{-3}$. Pearson correlation between source and target contrast maps is systematically and significantly increased when aligned using FUGW, as illustrated in Figure 3 where correlation grows by almost 40% from 0.258 to 0.356.

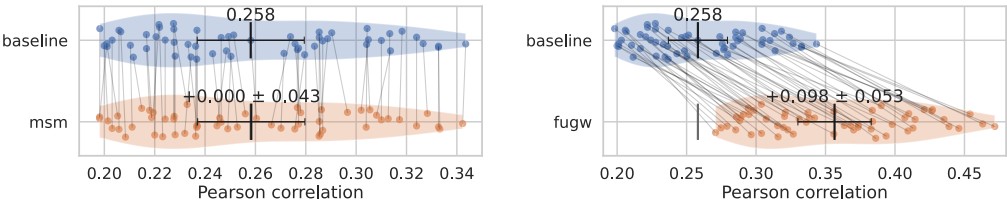

**Figure 3: Comparison of gains in correlation after inter-subject alignment** For each pair of source and target subjects of the dataset, we compute the average Pearson correlation between 30 test contrasts, leading to the (baseline) correspondence score, and compare it with that of the same contrast maps aligned with either MSM (left) or FUGW (right). Correlation gains are much better for FUGW.

We also varied training sets by selecting subsets of training contrasts and find that similar performance on the test set can be achieved regardless of the training data (see Supplementary section A.5 and in particular Supplementary Table S1).

**Hyper-parameters selection** Hyper-parameters used to obtain these results were chosen after running a grid search on $\alpha$, $\varepsilon$ and $\rho$ and evaluating it on the validation dataset. Computation took about 100 hours using 4 Tesla V100-DGXS-32GB GPUs. More precisely, it takes about 4 minutes to compute one coupling between a source and target 10k-vertex hemisphere on a single GPU, when the solver was set to run 10 BCD and 400 Sinkhorn iterations. In comparison, MSM takes about the same time on Intel(R) Xeon(R) CPU E5-2698 v4 @ 2.20GHz CPUs. Results are reported in Figure 4 and provide multiple insights concerning FUGW.

Firstly, without anatomical constraint ($\alpha = 0$), source vertices can be matched with target vertices that are arbitrarily far on the cortical sheet. Even though this can significantly increase correlation, it also results in very high vertex displacement values (up to $100mm$). Such couplings are not anatomically plausible. Secondly, without functional information ($\alpha = 1$), couplings recover a nearly flawless matching between source and target meshes, so that, when $\varepsilon = 10^{-5}$ (ie when we force couplings to find single-vertex-to-single-vertex matches), vertex displacement and spread are close to 0 and correlation is unchanged. Fusing both constraints ($0 < \alpha < 1$) yields the largest gains in correlation while allowing to compute anatomically plausible reorganizations the cortical sheet between subjects.

The impact of $\rho$ (controlling marginal penalizations) on correlation seems modest, with a slight tendency of increased correlation in unbalanced problems (low $\rho$).

Finally, it is worth noting that a relatively wide range of $\alpha$ and $\rho$ yield comparable gains. The fact that FUGW performance is weakly sensitive to hyper-parameters makes it a good off-the-shelf tool for neuroscientists who wish to derive inter-individual alignments. However, $\varepsilon$ is of dramatic importance

in computed results and should be chosen carefully. Vertex spread is a useful metric to choose sensible values of $\varepsilon$; for human data one might consider that it should not exceed $20mm$.

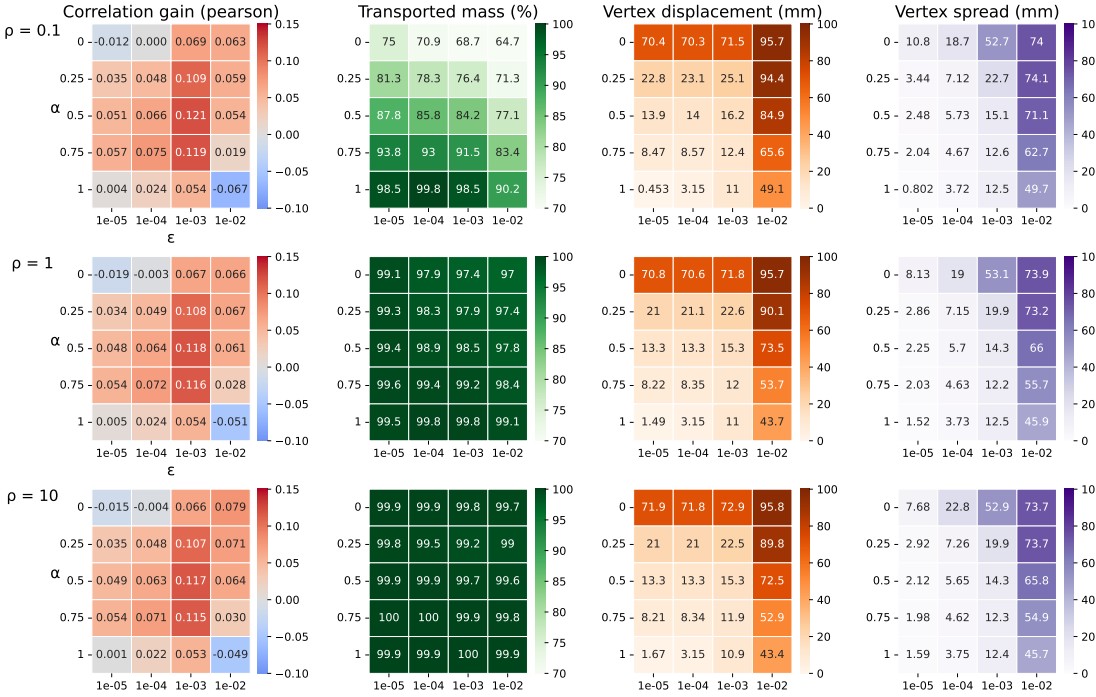

**Figure 4: Exploring hyper-parameter space to find relevant couplings** Given a transport plan aligning a source and target subject, we evaluate how much this coupling (left) improves correlation between unseen contrast maps of the two subjects, (center left) actually transports data, (center right) moves vertices far from their original location on the cortical surface and (right) spreads vertices on the cortical sheet. We seek plans that maximize correlation gain, while keeping spread and displacement low enough.

**Mass redistribution in unbalanced couplings** Unbalanced couplings provide additional information about how functional areas might differ in size between pairs of individuals. This is illustrated in Figure 5, where we observe variation in size of the auditory area between a given pair of individuals. This feature is indeed captured by the difference of mass between subjects (although the displayed contrast was not part of the training set).

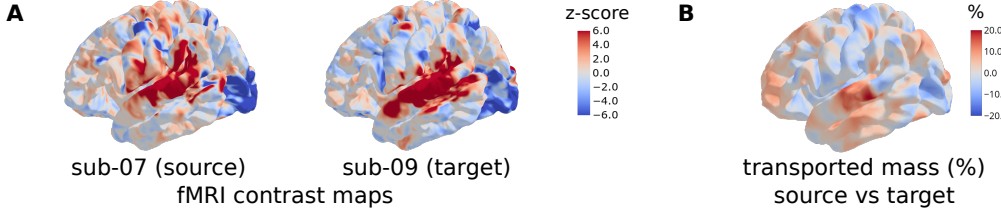

**Figure 5: Transported mass indicates areas which have to be resized between subjects** (Panel A) We show a contrast map from the test set which displays areas showing stronger activation during auditory tasks versus equivalent visual tasks. It shows much more anterior activations on the target subject compared to the source subject. This is consistent with the observation that more mass is present in anterior auditory areas of the source subject than in the target subject (Panel B).

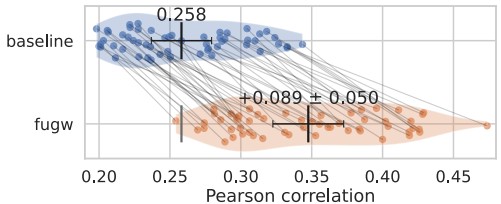

**Figure 6: Correlation between pairs of subjects is significantly better after alignment on individual anatomies than after projecting subjects onto a common anatomical template**

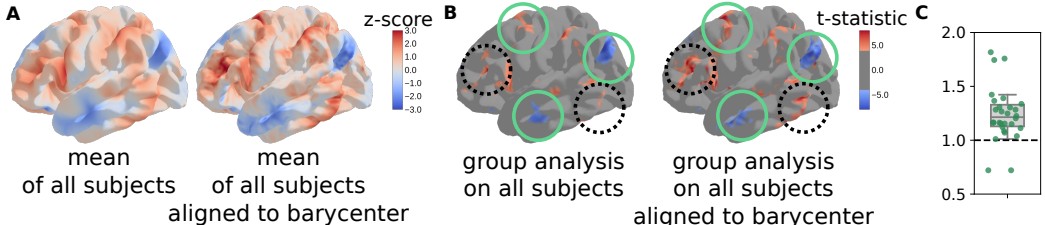

**Figure 7: FUGW barycenter yields much finer-grained maps than group averages** We study the same statistical map as in Figure 1, which contrasts areas of the brain involved in mathematical reasoning. **A**. These complex maps projected onto the barycenter and averaged show more specific activation patterns than simple group averages, especially in cortical areas exhibiting more variability, such as the prefrontal cortex. **B**. Deriving a t-test on aligned maps captures the same clusters as the classical approach (plain green circles), but also new clusters in areas where inter-subject variability is high (dotted black circles). Peak t-statistics are also higher with FUGW. **C**. Ratio of number of activated vertices ($|\text{t-statistic}| \geq 4$) with versus without alignment for each map of the test set. Our method finds significantly more of such vertices (p-value $= 3 \cdot 10^{-4}$).

## 4.2    Experiment 2 - Individual anatomies

As shown in Figure 6, we obtain correlation gains which are comparable to that of Experiment 1 (about 35% gain) while working on individual meshes. This tends to show that FUGW can compute meaningful alignments between pairs of individuals without the use of an anatomical template, which helps bridge most conceptual impediments listed in Section 1.

Moreover, this opens the way for computation of simple statistics in cohorts of individuals in the absence of a template. Indeed, one can pick an individual of the cohort and use it as a reference subject on which to transport all other individuals. We give an example in Figure S4, showing that FUGW correctly preserved idiosyncrasies of each subject while transporting their functional signal in an anatomically sound way.

## 4.3    Experiment 3 - Barycenter

In the absence of a proper metric to quantify the correctness of a barycenter, we first qualitatively compare the functional templates obtained with and without alignment. In Figure 7.A, we do so using brain maps taken from the test set. We can see that the barycenter obtained with FUGW yields sharper contrasts and more fine-grained details than the barycenter obtained by per-vertex averaging. We also display in Figure 7.B the result of a one-sample test for the same contrast, which can readily be used for inference. The one-sample test map obtained after alignment to the FUGW template exhibits the same supra-threshold clusters as the original approach, but also some additional spots which were likely lost due to inter-subject variability in the *fsaverage5* space. This approach is thus very useful to increase power in group inference. We quantify this result by counting the number of supra-threshold vertices with and without alignment for each contrast map of the test set. Our alignment method significantly finds more such vertices of interest, as shown in Figure 7.C.

# 5 Discussion

FUGW can derive meaningful couplings between pairs of subjects without the need of a pre-existing anatomical template. It is well-suited to computing barycenters of individuals, even for small cohorts.

In addition, we have shown clear evidence that FUGW yields gains that cannot be achieved by traditional diffeomorphic registration methods. These methods impose very strong constraints to the displacement field, that may prevent reaching optimal configurations. More deeply, this finding suggests that brain comparison ultimately requires lifting hard regularity constraints on the alignment models, and that two human brains differ by more than a simple continuous surface deformation. However, current results have not shown a strong correlation gain of unbalanced OT compared to balanced OT, likely because the cohort under study is too small. Leveraging datasets such as HCP [43] with a larger number of subjects will help lower the standard error on correlation gain estimates. In this work, we decided to rely on a predefined anatomical template (*fsaverage5*) to derive functional barycenters. It would be interesting to investigate whether more representative anatomical templates can be learned during the process. This would in particular help to customize templates to different populations or species. Additionally, using an entropic solver introduces a new hyper-parameter $\varepsilon$ that has a strong effect, but is hard to interpret. Future work may replace the scaling algorithm [11] used here by the majorization-minimization one [9], which does not require entropic smoothing. This solution can yield sparse couplings while being orders of magnitude faster, which will prove useful when computing barycenters on large cohorts.

Finally, we plan to make use of FUGW to derive alignments between human and non-human primates without anatomical priors. Indeed, the understanding of given brain mechanisms will benefit from more detailed invasive measurements made on other species *only if* brains can be matched across species; moreover, this raises the question of features that make the human brain unique, by identifying patterns that have no counterpart in other species. By maximizing the functional alignment between areas, but also allowing for some regions to be massively shrunk or downright absent in one species relative to the other, the present tool could shed an objective light on the important issue of whether and how the language-related areas of the human cortical sheet map onto the architecture of non-human primate brains.

## Acknowledgement

We thank Maëliss Jallais and Thomas Moreau for their help in implementing a Python wrapper for Multimodal Surface Matching, Thibault Séjourné and Gabriel Peyré for the useful discussions. We also thank Emma Robinson and Logan Williams for helping us understand better the ropes of MSM and how to tweak it's hyper parameters.

A.T, T. Z, S. D and B.T's research has received funding from the European Union's Horizon 2020 Framework Programme for Research and Innovation under the Specific Grant Agreement No. 945539 (Human Brain Project SGA3). It has also been supported by the the KARAIB AI chair (ANR-20-CHIA-0025-01) and the NeuroMind Inria associate team.

H. T, N. C and R. F's work is funded by the projects OTTOPIA ANR-20-CHIA-0030, 3IA Côte d'Azur Investments ANR19-P3IA-0002 of the French National Research Agency (ANR), the 3rd Programme d'Investissements d'Avenir ANR-18-EUR-0006-02, and the Chair "Business Analytics for Future Banking" sponsored by NATIXIS. This research is produced within the framework of Energy4Climate Interdisciplinary Center (E4C) of IP Paris and Ecole des Ponts ParisTech.

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
