# OpenReview forum: "Aligning individual brains with fused unbalanced Gromov Wasserstein"
_NeurIPS.cc/2022/Conference — NeurIPS 2022 Accept_

### Official Review · Reviewer_gBHr · 2022-07-03

**Rating:** 7
**Confidence:** 3
**Soundness:** 3 good
**Presentation:** 3 good
**Contribution:** 3 good

**Summary:**

A common problem in the use of brain imaging data is that individual brains differ in both geometry and function, requiring some form of alignment before comparison. The authors propose a novel method for aligning different brains, FUGW, which uses a combination of distance losses to respect functional, geometric, and even fundamental region differences between different brain images

**Questions:**

What would the impacts of these improvements be on downstream experiments/comparisons? There’s clearly a numerical advantage, but does that translate to application or understanding?

Do you have any examples of actually aligning brain images where one lacks certain region(s)? The toy sphere example sort of shows this


**Limitations:**

The technique may not scale well computationally, though that could more be an issue of the size of the images themselves.

**Strengths And Weaknesses:**

Strengths:

The problem statement and motivations are clear. The problem of accurate brain alignment is interesting both for current human neuroscience research and potential future cross-species comparisons.
The paper is well organized and clearly written. Toy examples are given to demonstrate the differences of their method
The method itself seems technically sound, with appropriate references to past work and a clear intuition of what each term does.
The numerical experiments are well-designed and explained, and they demonstrate a marked advantage of the proposed method in both quantitative and qualitative terms.

Weaknesses:
I would have liked to see the impact of these brain alignments on downstream experiments/tasks.
I would also be interested in computational comparisons; how much more difficult is it to do this vs MSM? In experiment 2, it was necessary to reduce the scale of the individual brain meshes to run at all (though admittedly, the other methods couldn’t even be applied)

---

> ### Author Response · Authors · 2022-08-02
> **Answer to Reviewer 3**
>
> Thank you for the time your spent reviewing this work and for your comments on the manuscript.
>
> > I would also be interested in computational comparisons; how much more difficult is it to do this vs MSM?
>
> - Timings for MSM were added line 219. Generally speaking, depending on the configuration we use, timings for MSM could vary from ~5 minutes to ~30 minutes when aligning two 10k meshes.
>
> > Questions:
> >What would the impacts of these improvements be on downstream experiments/comparisons? There’s clearly a numerical advantage, but does that translate to application or understanding?
>
> - To address your point, we have added the two following experiments to the supplementary material (see section A5 of the supplementary material), based on additional data that were available on the IBC dataset, namely T1 / T2 ratio and movie watching:
>     - we compare the correlation of myelin maps (inferred by computing the T1 / T2 ratio in individual subjects) before and after they have been aligned using fMRI data. Correlation gain does not seem significant, which we believe can be partially explained by the fact that myelin maps are already very consistent across individuals as most vertices showing high-value are located in areas that are very stable across individuals in freesurfer space (see Figure S7).
>     - we used movie-watching data to compute OT plans and used the latter to transport the same fMRI contrast maps used in Experiments 1, 2 and 3. These data have little in common with the test set and yet we find significant correlation gains.
> Overall this suggests that the estimated transport plans are beneficial for functional features, not so much for anatomical ones.  Additional experiments are needed with e.g. connectivity data.
>
>
> > Do you have any examples of actually aligning brain images where one lacks certain region(s)? The toy sphere example sort of shows this
>
> - Empirically, what we observe is that lower values of $\rho$ yield lower masses everywhere on the cortex, and we could not exhibit an alignment which completely discards brain regions of the source or target subject.
> - This could come from the fact that our mass contraint uses the Kullback-Leibler divergence, which eventually smoothes mass on the mesh. Other divergences (Total Variation for instance) might yield sparser mass distributions (see Figures 1-4 of [1] to get a better intuition concerning this idea)
> - In general, we believe the human-to-monkey alignment problem will be more suited to studying these effects.
>
> > Limitations:
> > The technique may not scale well computationally, though that could more be an issue of the size of the images themselves.
>
> - Indeed, this current implementation does not scale well. Here are some of the ideas we would like to test in a near future to enhance the solver:
>     1. For low values of $\epsilon$ (which is what we are interested in), the plan should be mostly sparse. Using a coarse-to-fine strategy to approximate sparse solutions should greatly speedup the process.
>     2. Reduce distance matrices with kernel operations using LMDS embedding and combine it with a library to efficiently deal with kernel matrices like KeOps (https://github.com/getkeops/keops)
>
> [1] Séjourné, Thibault, Jean Feydy, François-Xavier Vialard, Alain Trouvé, and Gabriel Peyré. ‘Sinkhorn Divergences for Unbalanced Optimal Transport’. arXiv, 19 March 2021. https://doi.org/10.48550/arXiv.1910.12958.

---

### Official Review · Reviewer_SBLC · 2022-07-09

**Rating:** 7
**Confidence:** 4
**Soundness:** 4 excellent
**Presentation:** 4 excellent
**Contribution:** 3 good

**Summary:**

The authors propose an optimal transport based registration method for meshes with multi-variate features, a principle use-case of which is neuro-imaging (cortical mesh alignment). They construct a "Gromov-Wasserstein" loss which is the usual transport distance term to penalize large deformations w.r.t. the mesh geodesic, which they use alongside a "Wasserstein" loss, which is the transport matching term. Overall the transport method is defined for an "unbalanced" transport problem, where the domains need not match and where restrictions may be placed on transport to/from specific regions.

Experiments are presented with both synthetic and real data, using subsets of the Individual Brain Charting dataset (composed of a variety of task fMRI and the standard anatomical scans). Comparisons are made with Multisurface Matching (MSM) (Robinson 2014), as well as a template analysis (barycenters).

**Questions:**

My primary concern is with evaluation. While evaluating non-linear inter-subject registrations is always difficult, here we have chosen to ignore landmarks and focus on functional correlation of a hold-out set. While this is not unreasonable on its own, it is in the optimization desiderata themselves, and also in the hyper-parameter tuning objective. How can we know that these registrations are reasonable? Hold out correlation is one solution, but surely these trials are not fully independent (they do come from the same heads and, importantly, the same arterial structures). Could these comparisons be made using, e.g., myelin patterning, or its proxy by T1/T2 ratio?

Another issue is that the comparison is made against MSM, which has default parameters set for an HCP-like dataset. Were the same search-grid to be made for MSM, would we see a similar increase? It appears to do nothing to the correlations; perhaps this is actually the case, but it seems like a very weak baseline.

The authors specifically avoid diffeomorphic constraints. Given this, and average displacement on the order of 1-2cm (Fig 4), do we find extreme topological changes? Extreme geometric changes? Sulci to gyri?

Other comments:
* Robinson 2014 is not cited in the actual text of the paper; it may be helpful to readers during the introduction of MSM to do so.
* Very little time is spent on the solution method, or the mesh-reduction method. While understandably space is constrained, I feel it would be helpful to have more explanation in the main text.

**Limitations:**

None.

**Strengths And Weaknesses:**

Strengths:
* This paper is well written, and enjoyable to read.
* The treatment of the transport problem is well done, and the implementation of a standard solution method (scaling) to this particular problem seems sound.

Weaknesses:
* The method requires a dense interaction matrix of $V^2$ size, where $V$ is the number of vertices on the mesh, limiting this method to something on the order of 10k vertices. (Though, to the authors credit, considerations were made here.)
* The baseline method is untuned.

Comment:
* The technical innovation is limited to a specific application; this is not disqualifying, but it is limiting in "significance" to the NeurIPS community relative to, say, a conference or journal from the application domain (e.g., IEEE TMI).

---

> ### Author Response · Authors · 2022-08-02
> **Answer to Reviewer 2**
>
> Thank you for the time your spent reviewing this work and for your comments on the manuscript.
>
> > While evaluating non-linear inter-subject registrations is always difficult, here we have chosen to ignore landmarks and focus on functional correlation of a hold-out set. While this is not unreasonable on its own, it is in the optimization desiderata themselves, and also in the hyper-parameter tuning objective. How can we know that these registrations are reasonable? [...] Could these comparisons be made using, e.g., myelin patterning, or its proxy by T1/T2 ratio?
>
> - To address your point, we have added the two following experiments to the supplementary material (see section A5 of the supplementary material), based on additional data that were available on the IBC dataset, namely T1 / T2 ratio and movie watching:
>     - we compare the correlation of myelin maps (inferred by computing the T1 / T2 ratio in individual subjects) before and after they have been aligned using fMRI data. Correlation gain does not seem significant, which we believe can be partially explained by the fact that myelin maps are already very consistent across individuals as most vertices showing high-value are located in areas that are very stable across individuals in freesurfer space (see Figure S7).
>     - we used movie-watching data to compute OT plans and used the latter to transport the same fMRI contrast maps used in Experiments 1, 2 and 3. These data have little in common with the test set and yet we find significant correlation gains.
> Overall this suggests that the estimated transport plans are beneficial for functional features, not so much for anatomical ones.  Additional experiments are needed with e.g. connectivity data.
>
>
> > Another issue is that the comparison is made against MSM, which has default parameters set for an HCP-like dataset. Were the same search-grid to be made for MSM, would we see a similar increase? It appears to do nothing to the correlations; perhaps this is actually the case, but it seems like a very weak baseline.
>
> - It is hard to grid-search hyper-parameters of MSM since there are a lot of them (here is a typical config file: https://github.com/ecr05/MSM_HOCR/blob/master/config/basic_configs/config_standard_MSMpair) and they could/should have different values for each of the MSM block of iterations (usually between 4 and 5 blocks ; see `--it` in the previously linked file). After exchanging verbally with people who developped this method, we had narrowed down our search to only one parameter (`--lambda` in the previous config file), which controls the regularity of computed alignments at each block of iterations, and had run a cross-validated gridsearch similar to what we did for FUGW.
> - We noticed that default config files were recently updated on https://github.com/ecr05/MSM_HOCR, so we ran a new cross-validated gridsearch with these params to address your concern, but new results were very similar to previous ones.
>
> > The authors specifically avoid diffeomorphic constraints. Given this, and average displacement on the order of 1-2cm (Fig 4), do we find extreme topological changes? Extreme geometric changes? Sulci to gyri?
>
> - Empirically, we can indeed exhibit plans that contain matchings between source sulci and target gyri (and vice-versa). However, we advocate that data used to derive these plans also show patterns of activation which, across subjects, are located inconsistently in gyri or sulci. Maybe this problem could be accounted for by including sulcal depth in the alignment features.
>
> > Other comments:
> >
> > Robinson 2014 is not cited in the actual text of the paper; it may be helpful to readers during the introduction of MSM to do so.
>
> - This piece of work is cited once (see ref 25, line 43 of the old paper). To help readers, we repeated this citation and added one to Robinson 2018 - an extension of Robinson 2014 - line 141 of our revised paper (refs 34 and 35 of the new paper).
>
> > Very little time is spent on the solution method, or the mesh-reduction method. While understandably space is constrained, I feel it would be helpful to have more explanation in the main text.
>
> - We addressed your point by:
>     - adding details to the description of FUGW (lines 109-116 of the revised paper) in the main text as we thought this was the most important part
>     - moving the mesh-reduction method explanation (previously described at lines 187-194) to the supplementary material. We rephrased it more extensively there.

---

### Official Review · Reviewer_J9Ws · 2022-07-16

**Rating:** 6
**Confidence:** 5
**Soundness:** 3 good
**Presentation:** 3 good
**Contribution:** 3 good

**Summary:**

The manuscript proposed an unsupervised image registration algorithm based on the optimal transfer scheme, solving an unbalanced fused Gromov-Wasserstein optimization problem. The algorithm was evaluated on an fMRI data registration task to test whether the correspondence between fMRI signals can be improved after the registration.

**Questions:**

The proposed FUGW framework was tested on a cortical surface registration task on the activation maps and evaluated by the correlation of fMRI signals before/after the alignment. It will be great if it can be evaluated by anatomical features of the cortical surface (e.g., correspondence between the major gyri/sulci locations), which can be done by a small amount of human annotation. This is specifically important as in the paper, the fMRI-derived activation value is used as the feature to perform the registration, then the registration was evaluated by the same fMRI signal’s correlation. Thus, the improved correlation after the registration is somehow expected, as the vertex with a similar response to stimuli (reflected in the activation value) would usually have a similar fMRI signal.

As the author mentioned in line 220, the selection of α is important for balancing anatomical validity and registration effectiveness. Is there a way to estimate this hyperparameter in a more intuitive way (e.g., by enforcing voxels in the same gyri/sulci would not move to another gyri/sulci)?

More details of the FUGW framework need to be clarified to understand its mechanism and apply it for other tasks:
1)	 How w^s and w^t in Eq. 1 are defined? The author mentioned that “We also assign the distribution w^s∈R^n_+ on the source vertices, which we interpret as their relative importance in the mesh” but more details are needed.
2)	For the Barycenter estimation, how the mesh of the Barycenter is defined/updated (e.g., number of vertices) in each iteration? As each subject has a different number of vertices, it is curious that whether the Barycenter has a fixed number of vertices or varying vertices at each iteration.

In section 4.1, the first experiment, the author mentioned “Aligning subjects on a fixed mesh”, does the “fixed mesh” here indicate the target of registration which is the template?

**Limitations:**

N/A, this works only developed an image registration algorithm.

**Strengths And Weaknesses:**

Effective registration from one individual image to a common space, as well as from one individual to another individual, is highly needed in medical image analysis. Although various registration algorithms have been developed, the proposed fused Gromov-Wasserstein optimal transfer algorithm with unbalanced matching is a novel and plausible way to solve this task in an unsupervised approach. The capability of considering both the image itself (i.e., the mesh) and features defined on the image (e.g., the activation map used in this work) is much needed especially for the multi-modal image fusion. The methodology of this work is clearly written, and the overall quality of this manuscript is good.

The major issue of this work is that its evaluation scheme needs to be improved, see questions below.

---

> ### Author Response · Authors · 2022-08-02
> **Answer to Reviewer 1**
>
> Thank you for the time your spent reviewing this work and for your comments on the manuscript.
>
> > The proposed FUGW framework was tested on a cortical surface registration task on the activation maps and evaluated by the correlation of fMRI signals before/after the alignment. It will be great if it can be evaluated by anatomical features of the cortical surface (e.g., correspondence between the major gyri/sulci locations), which can be done by a small amount of human annotation. This is specifically important as in the paper, the fMRI-derived activation value is used as the feature to perform the registration, then the registration was evaluated by the same fMRI signal’s correlation. Thus, the improved correlation after the registration is somehow expected, as the vertex with a similar response to stimuli (reflected in the activation value) would usually have a similar fMRI signal.
>
> - To address your point, we have added the two following experiments to the supplementary material (see section A5 of the supplementary material), based on additional data that were available on the IBC dataset, namely T1 / T2 ratio and movie watching:
>     - we compare the correlation of myelin maps (inferred by computing the T1 / T2 ratio in individual subjects) before and after they have been aligned using fMRI data. Correlation gain does not seem significant, which we believe can be partially explained by the fact that myelin maps are already very consistent across individuals as most vertices showing high-value are located in areas that are very stable across individuals in freesurfer space (see Figure S7).
>     - we used movie-watching data to compute OT plans and used the latter to transport the same fMRI contrast maps used in Experiments 1, 2 and 3. These data have little in common with the test set and yet we find significant correlation gains.
> Overall this suggests that the estimated transport plans are beneficial for functional features, not so much for anatomical ones.  Additional experiments are needed with e.g. connectivity data.
>
> > As the author mentioned in line 220, the selection of α is important for balancing anatomical validity and registration effectiveness. Is there a way to estimate this hyperparameter in a more intuitive way (e.g., by enforcing voxels in the same gyri/sulci would not move to another gyri/sulci)?
>
> - We observe that changing values of alpha (as long as 0 < $\alpha$ < 1) does not have a dramatic impact on computed transport plans (see Figure 4), so we believe a default $\alpha=0.5$ is a safe default. However, $\epsilon$ needs to be set carefully, which is why we are working to implement a solver which directly estimates FUGW (rather than via its entropic regularization, as currently used), following work done in [1] for instance.
>
> > More details of the FUGW framework need to be clarified to understand its mechanism and apply it for other tasks: 1) How w^s and w^t in Eq. 1 are defined? The author mentioned that “We also assign the distribution w^s∈R^n_+ on the source vertices, which we interpret as their relative importance in the mesh” but more details are needed.
>
> - "relative importance in the mesh" was an ambiguous formulation. We removed it.
> - $w \in \mathbb R^n_+$ is a discrete distribution (not necessarily a probability) of the vertices. Here, $w$ represents the mass that can be transported from (resp. to) each vertex of the source (resp. target) subject. Since, we do not know which areas of the cortex should have more mass than others in a given subject, we use uniform distributions over the cortical surface in all our experiments (see line 85 of the old paper), i.e. $w^s = \frac{1}{n} 1_n$ and $w^t = \frac{1}{p} 1_p$ (see lines 85-86 of the revised paper).
>
> > 2) For the Barycenter estimation, how the mesh of the Barycenter is defined/updated (e.g., number of vertices) in each iteration? As each subject has a different number of vertices, it is curious that whether the Barycenter has a fixed number of vertices or varying vertices at each iteration.
>
> - In the current implementation, the number of vertices in the barycenter mesh is set by the user at initialisation and stays fixed during the block coordinate descent (BCD) scheme. At each iteration, we update $D^B$, the matrix of pairwise anatomical distances between barycenter vertices (see section A3 in the Appendix for more details). After the scheme finishes, we can threshold $D^B$ to select edges of the computed barycenter mesh.
>
> > In section 4.1, the first experiment, the author mentioned “Aligning subjects on a fixed mesh”, does the “fixed mesh” here indicate the target of registration which is the template?
>
> - Indeed, "fixed mesh" means we set the mesh to fsaverage5 (see lines 195-198 of the old paper).
> - This has been made more explicit, see lines 193-197 of the revised paper.
>
> [1] Chapel, Laetitia, et al. ‘Unbalanced Optimal Transport through Non-Negative Penalized Linear Regression’. http://arxiv.org/abs/2106.04145.

---

### Meta-Review · Area_Chair_V2mi · 2022-08-24

**Recommendation:** Accept
**Confidence:** Certain

**Metareview:**

This paper uses optimal transport for aligning cortical surfaces based on the similarity of their functional signatures under different stimulations. The paper is well written, and the experimental setup is sound. The authors added experiments and clarifications to address reviewers' comments and concerns. The reviewers provided a consensus accept rating for this paper.


**Award:**

No

---

### Decision · Program_Chairs · 2022-09-14

Accept